# Left Atrial Remodeling and Brain Natriuretic Peptide Levels Variation after Left Atrial Appendage Occlusion

**DOI:** 10.3390/jcm10153443

**Published:** 2021-08-03

**Authors:** Thibaut Pommier, Thibault Leclercq, Charles Guenancia, Carole Richard, Guillaume Porot, Gabriel Laurent, Luc Lorgis

**Affiliations:** 1Department of Cardiology, University Hospital, 21000 Dijon, France; thibault.leclercq@chu-dijon.fr (T.L.); charles.guenancia@chu-dijon.fr (C.G.); carole.richard@chu-dijon.fr (C.R.); guillaume.porot@chu-dijon.fr (G.P.); gabriel.laurent@chu-dijon.fr (G.L.); luc.lorgis@chu-dijon.fr (L.L.); 2Laboratory of Cerebro-Vascular Pathophysiology and Epidemiology (PEC2), University of Burgundy, 21000 Dijon, France

**Keywords:** left atrial appendage occlusion, bleeding risk, BNP, atrial remodeling, atrial fibrillation, atrial cardiopathy

## Abstract

Background: Few data are available about brain natriuretic peptide (BNP) variation and left atrial remodeling after the left atrial appendage occlusion (LAAO) technique. Methods: Prospective study included all consecutive patients successfully implanted with an LAAO device. Contrast-enhanced cardiac computed tomography (CT) was performed before and 6 weeks after the procedure with reverse left atrial remodeling defined by an increase in LA volume >10%, together with blood sampling obtained before, 48 h after device implantation and at the first visit after discharge (30–45 days) for BNP measurement. Results: Among the 43 patients implanted with a complete dataset, mean end-diastolic LA volume was 139 ± 64 mL and 141 ± 62 mL at baseline and during follow-up (45 ± 15 days), respectively, showing no statistical difference (*p* = 0.45). No thrombus was seen on the atrial side of the device. Peridevice leaks (defined as presence of dye in the LAA beyond the device) were observed in 17 patients (40%) but were trivial or mild. Reverse atrial remodeling (RAR) at 6 weeks was observed in six patients (14%). Despite no difference in BNP levels on admission, median BNP levels at 48 h were slightly increased in RAR patients when compared with controls. During FU, BNP levels were strictly identical in both groups. These results were not modified even when each RAR case was matched with two controls on age, LVEF, creatinine levels and ACE inhibitors treatment to avoid potential confounders. Conclusion: Our study showed that despite the fact that the LAAO technique can induce left atrial remodeling measured by a CT scan, it does not seem to impact BNP levels on the follow-up. The results need to be transposed to clinical outcomes of this expanding population in future studies.

## 1. Introduction

Atrial fibrillation (AF) is the most common heart rhythm disorder and the second most common cause of stroke [1]. Atrial fibrillation prevalence increases with age, and the current demographic evolution with an elderly population makes it a major cause of morbidity [2]. Thrombus that emerges inside the left atrial appendage (LAA) is the most common cause of ischemic stroke in non-valvular atrial fibrillation patients [3]. Anticoagulation effectively reduces the risk of ischemic stroke and all-cause death in patients with AF and is still currently the reference treatment of thromboembolic risk prevention in patients with atrial fibrillation but sometimes represents an inappropriate option for some patients [4]. Firstly, anticoagulation also increases the risk of hemorrhage complications in patients at high risk of bleeding, and several studies confirm that the AF population is under-treated with nearly 40% of patients at risk for stroke who do not receive any form of oral anticoagulation [5]. Moreover, approximately 10% of the patients have a contraindication to anticoagulation, and 2% have an absolute contraindication.

In patients with non-valvular AF, percutaneous left atrial appendage occlusion (LAAO) has emerged as an alternative approach to reduce the risk of stroke, especially in patients with high bleeding risk or a contraindication of oral anticoagulation [6]. LAA is a frequent source of systemic embolism due to blood stasis in the LAA in the fibrillating atrium leading to thrombus formation, and it is the reason why an occlusion or removal of the LAA may decrease the risk of systemic embolization. Nearly 92% of LA thrombi are localized in the LAA [7]. There are several methods of LAA closure, including percutaneous [8] and surgical approaches [9]. Concerning the interventional approach, left atrial appendage occlusion is performed with the implantation of a device excluding the LAA [10,11]. Nowadays, according to ESC Guidelines, LAAO may be considered for stroke prevention in patients with AF and contraindications for long-term anticoagulant treatment (those with a previous life-threatening bleeding without a reversible cause) [12].

LAAO is therefore effective to reduce the risk of stroke in well-selected AF patients. However, few data are available about the potential hemodynamic consequences of left atrial appendage occlusion. The role of the LAA in cardiac hemodynamics has been previously investigated in both animal and human studies [7]. LAA is a complex structure with variable shape and size and with effective contractions during sinus rhythm, but contractions disappear during AF, leading to the formation of thrombi. Left atrial appendage has several important mechanical and endocrine functions. It is a reservoir chamber due to its distensible ability, allowing a response to an increase in the volume or the overload of the left atrium [13]; an exclusion of the LAA may encourage an increase in the volume of the left atrium. Then, the LAA has a neurohormonal activity and is known as the source of the atrial natriuretic peptide (ANP) with secretion in response to an increased atrial volume or pressure, allowing a vasodilator and diuretic activity and a decrease of blood pressure. Studies found that patients with LAAO have significantly lower ANP secretion and concomitant increase of cardiac congestion.

Therefore, LAA functions will be changed after LAAO. Consequently, these modifications can affect cardiac function and structure, but there are few data about the impact of percutaneous LAAC on left atrial functional and structural remodeling.

In our center, the first patients who had undergone an LAA occlusion experienced a rise in the brain natriuretic peptide (BNP) associated with cardiac congestion. Nevertheless, even if the relationship between the ANP and the LAA is established, there are limited data about BNP variation and left atrium remodeling after the left atrial appendage occlusion technique. The objectives of our study were to assess the relationship between BNP variation and left atrium remodeling after the left appendage occlusion (LAAO) technique.

## 2. Materials and Methods

### 2.1. Study Flow Chart

All consecutive patients successfully implanted with an LAAO device using either the Amplatzer Cardiac Plug (ACP) device (St. Jude Medical, Minneapolis, MN, USA) or the Watchman device (Boston Scientific, Natick, MA, USA) were included in a prospective single-center study at Dijon University Hospital for a period of four years. A complete screening with all clinical and paraclinical signs of the implanted patients was noted before the procedure, just like the data concerning the percutaneous intervention. A contrast-enhanced cardiac computed tomography (CT) was performed before and 6 weeks after the procedure with reverse left atrial remodeling defined by an increase in LA volume >10%. A blood sampling was obtained before, 48 h after device implantation and at the first visit after discharge (30–45 days) for BNP measurement. Major adverse cardiac events (MACEs) were collected after device implantation and during the follow-up.

After 6 weeks, patients were classified according to the presence or absence of left atrial remodeling, based on the second CT evaluation.

The study sample consisted of 43 patients, with at least one-year follow-up. The flow chart is shown in Figure 1.

### 2.2. Cardiac CT Analysis

All cardiac CT images were analyzed by experienced practitioners using syngo.via software. An automated 3D region-growing segmentation algorithm was used to calculate the volumes of the left atrium [14] (Figure 2). Left atrial volumes were estimated by two experienced practitioners because of the novelty of the technique, and there was a difference of less than 5% between the two measurements proving the reproducibility of the technique. CT was performed pre-procedurally and 6 weeks post-procedurally to evaluate left atrial volumes but also device position, device thrombus or the presence of residual peridevice leaks.

### 2.3. Statistical Analysis

Statistical analyses were performed using SPSS version 12.0.1 (Statistical Package for the Social Sciences, IBM Inc., Armonk, NY, USA). Dichotomous variables were expressed as *n* (%) and continuous variables as mean and standard deviation. A Kolmogorov–Smirnov test was performed to analyze the normality of continuous variables. Mann–Whitney test (skewed data) or Student’s *t*-test (unskewed data) was used to compare continuous data, and the Chi 2 test or Fisher’s test was used for dichotomous data. Significance threshold was set at 5%.

## 3. Results

### 3.1. Patients Baseline Characteristics

Baseline characteristics on admission to the hospital are summarized in Table 1.

The study included 43 patients, with a median age of 75, with a male predominance (65%). In the implanted patients, the median CHA_2_DS_2_-VASc score was nearly 4 while the median HAS-BLED score was also nearly 4. Most patients had permanent atrial fibrillation (70%), and most patients had a prior stroke (70%) in the medical history. Eighty-four percent of the patients reported a prior bleeding, most of whom had normal left ventricular function.

In the cohort, four indications for LAAO were revealed: previous bleeding under OAC (84%), cerebral amyloid angiopathy (12%), high risk of bleeding (2%) or stroke despite OAC (2%).

### 3.2. CT Results and Left Atrial Remodeling

The baseline CT examination was performed less than a month before the procedure, and the second CT was performed 48 ± 36 days after the device implantation. Results are shown in Table 2. At 6 weeks, 6 patients (14%) exhibited left atrial remodeling on CT, while the remaining 37 patients (86%) did not have a significant increase in left atrial volume. There were no statistically significant differences between the two groups according to left atrial remodeling in age, sex, demographic data or symptoms. The HAS-BLED score and the CHA_2_DS_2_-VASc score were the same in the two groups, the same as atrial fibrillation class, prior stroke or prior bleeding.

In the global population, mean LA volume was respectively 139 ± 64 mL at baseline and 140 ± 63 mL during follow-up, showing no statistical difference. Interestingly, reverse left atrial remodeling was found in six patients, with a median volume increase of 26 mL. No thrombus was found on the atrial side of the device. Peridevice leaks (defined as presence of dye in the LAA beyond the device) were observed in four patients, with no statistical difference between both groups (adverse remodeling or not).

### 3.3. Procedural and Post-Procedural Characteristics

Interventional LAA closure was successful in all the patients, and no procedure-related major complications were observed. At one year, six patients had died. Other complications included major bleeding (four patients) and congestive heart failure (eight patients). There was no significant difference between the two groups regarding procedural and post-procedural characteristics and results (Table 2).

### 3.4. Atrial Remodeling and Natriuretic Peptides

Blood samples including natriuretic peptides (B-type natriuretic peptide (BNP)) were collected according to a standardized method before the procedure, 48 h after the procedure and at distance. Firstly, the concentration of the BNP is not different between the two groups according to the presence or not of adverse remodeling of the left atrium, whatever the time of blood sampling. The results are summarized in Table 3. However, we noted that the patients with RAR had a trend of BNP increasing just after the device implantation, then stabilizing, while in those without RAR, a decrease in the BNP level after LAAO was observed (Figure 3). This correlation between RAR and increase of the BNP level after LAAO could be explained by the loss of reservoir function of the left atrium after LAAO, before an adaptation.

## 4. Discussion

In this study, we aimed to evaluate the impact of LAAO on cardiac hemodynamics based on imaging parameters using LA volume on cardiac CT over time.

We therefore noted that a proportion of patients had left atrial remodeling after an LAAO, even if this was clearly just a trend without statistical difference. However, the presence of cardiac remodeling associated with symptoms suggests that LAA removal can potentially induce heart failure. Obviously, this is the loss of the reservoir function of the LAA after closure which can explain post-procedural LA remodeling.

Percutaneous LAAO is sometimes the only option for ischemic stroke prevention, especially in patients with contraindication for long-term OAC or high risk of bleeding. Although the safety and efficacy of this treatment have been demonstrated in randomized controlled trials and real-world experience [15], data about the impacts of LAAO on cardiac remodeling are not well established with a small number of studies, and the present study brings additional data on this topic.

### 4.1. Left Atrial Remodeling after LAAO

In 2017, Jalal et al. [16] were the first to non-invasively evaluate the left atrial hemodynamic impact of percutaneous LAA closure and showed no early significant LA remodeling but a trend toward an increase in LV filling pressure. In their population of 63 patients, mean LA volume was 145 ± 55 mL and 144 ± 50 mL at baseline and during follow-up, respectively, without statistical difference. However, regarding echocardiographic assessment of diastolic function, a significant difference was observed in the E/E’ ratio which increased after LAA closure.

In the same line, Tan Phan et al. [17] showed in 2018 that there was a significant increase in LA size and LV filling pressure (also significant increase in the E/e’ ratio) among NVAF patients after LAAO. The authors concluded that the impact of LAA closure on cardiac functional and structural remodeling observed on echocardiography may have potential clinical implications.

Finally, Luani et al. [18] observed once again in cardiac echography that left atrial volume increased significantly after interventional LAA closure, but left atrial enlargement was not correlated with clinical progression of heart failure in this study.

Our study is in agreement with previous studies highlighting an increase in LA size in cardiac CT in some patients, as well as variations in natriuretic peptide levels, even if no significant difference or correlation was observed. This new study confirms the real potential for left atrial remodeling after LAAO, in agreement with previous studies. These hypotheses will have to be confirmed with future randomized controlled trials with regular follow-up by CT scan and also by echocardiography.

### 4.2. Natriuretic Peptide Variation after LAAC

The LAA allows production of the atrial natriuretic peptide (ANP) [19], which plays an important role in natriuresis and regulation of heart-filling pressures. With the emergence of LAA occlusion, we can expect variations in the levels of natriuretic peptides and thus changes in cardiac physiology. Unfortunately, although the LAA is a source of the ANP, the impact of LAAO on natriuretic peptide levels remains largely unknown. In his study, Lakkireddy [20] compared the levels of natriuretic peptides before LAAO then during the follow-up with several measurements for three months. In comparison with pre-endocardial LAA device levels, the concentration of the ANP was significantly higher immediately after the procedure. Twenty-four hours after the procedure and at 3 months, the levels of the ANP were not significantly different when compared with the pre-endocardial LAA device baseline. BNP levels displayed similar changes.

In our study, ANP was not available, and we therefore opted for the BNP assay in blood sampling to search for a link between left atrial remodeling and heart failure. Having data on the ANP would have been better.

Moreover, the role of natriuretic peptides is not only about cardiac congestion. ANP and BNP act on the heart, blood vessels and also on the kidneys. The natriuretic peptides have an important role in regulating the circulation. LAAO can therefore also cause changes in the levels of natriuretic peptides not linked to heart failure but potentially linked to vascular and renal changes, in particular, linked to changes in volemia after procedures such as LAAO, where the patients may benefit from vascular replenishment for hypotension or alternatively diuretics for congestion, with a variable intensity of production of natriuretic peptides according to the situation.

### 4.3. Limitations of the Study

The study was retrospective with a limited number of patients. It would have been interesting to have in addition an echocardiography follow-up and another measurement of the left atrial size and also an evaluation of the filling pressures. Furthermore, changes in medication that may affect cardiac remodeling were not evaluated. A limitation of this study is the lack of dosage of the ANP, a specific marker of the left atrial, produced by the atria, which was not available at the start of the study on the blood sampling of the patients.

The present study provided, however, new insights regarding the interaction between natriuretic peptides, atrial remodeling and left appendage occlusion and had the advantage to benefit a short-time follow-up, with many cardiac CTs in association with a clinical and biological follow-up.

## 5. Conclusions

Our study showed a trend in favor of remodeling of the left atrial measured by CT scan after an LAAO, just like some modifications in the BNP level around the device implantation without statistical difference with a low number of patients. The potential impact of LAAO concerning cardiac remodeling is, however, real. The results need to be transposed to clinical outcomes of this expanding population in future studies.

## Figures and Tables

**Figure 1 jcm-10-03443-f001:**
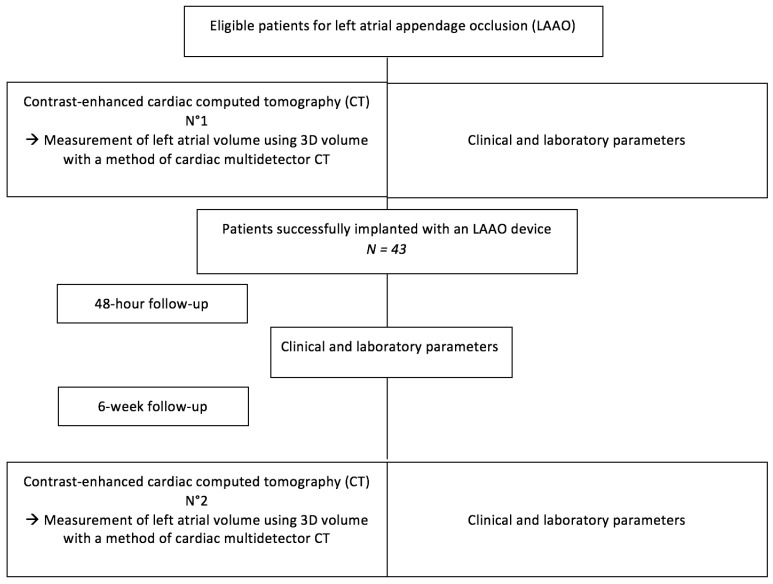
Flow chart of the study.

**Figure 2 jcm-10-03443-f002:**
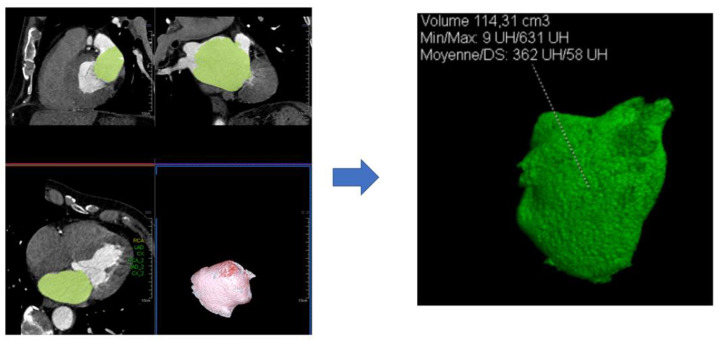
Left Atrial volume assessment using ECG-gated cardiac multidetector CT.

**Figure 3 jcm-10-03443-f003:**
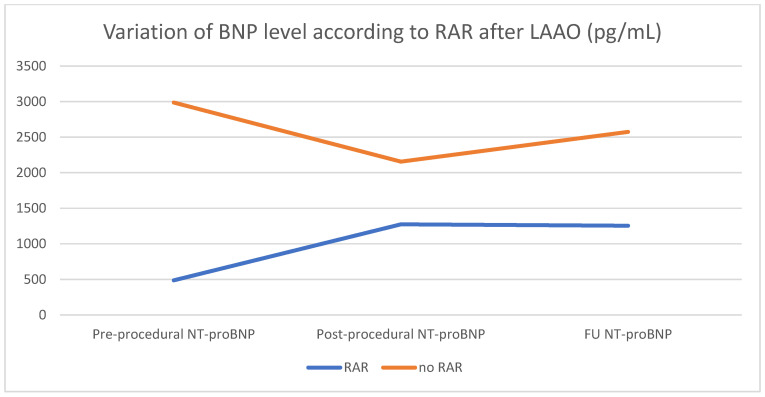
Variation of BNP level according to RAR after LAAO (pg/mL).

**Table 1 jcm-10-03443-t001:** Patients’ characteristics (according to the presence or absence of reverse atrial remodeling).

	RAR Group *N* = 6	No RAR Group *N* = 37	*p*
Age, years	80 ± 5.7	75 ± 8.2	0.120
Sex (male)	5 (84)	23 (62)	0.304
Hypertension	5 (84)	36 (98)	0.262
Diabetes	3 (50)	8 (22)	0.164
CHA_2_DS_2_-VASc score	4.3 ± 1.21	4.2 ± 1.4	0.873
HAS BLED score	4 ± 0.9	4.3 ± 1.26	0.498
Paroxysmal AF	1 (16)	12 (33)	0.401
Permanent AF	5 (84)	25 (67)	0.401
Prior stroke	5 (84)	25 (67)	0.401
Congestive heart failure	1 (16)	6 (16)	0.681
History of CAD	1 (16)	10 (27)	0.512
History of CABG	0 (0)	4 (11)	0.535
Prior bleeding	5 (84)	31 (84)	0.681
Prior TAVR	0 (0)	1 (3)	0.857
LVEF, %	60 ± 4	56 ± 9	0.110
Creatinine clearance, mL/min	66 ± 14	66 ± 26	0.919
Pre-procedural NT-proBNP, pg/mL	486 ± 392	2987 ± 6766	0.054
Post-procedural NT-proBNP, pg/mL	1274 ± 1316	2156 ± 3248	0.402
FU NT-proBNP, pg/mL	1255 ± 1603	2574 ± 3309	0.308
Post-procedural troponin, pg/mL	0.26 ± 0.17	0.34 ± 0.39	0.370
Indication for LAAO			
Previous bleeding under OAC	5 (84)	31 (84)	0.681
Cerebral amyloid angiopathy	1 (16)	4 (11)	
High risk of bleeding	0 (0)	1 (3)	
Stroke despite OAC	0 (0)	1 (3)	

Data are presented as *n* (%) and mean ± SD. AF, Atrial Fibrillation; CAD, Coronary Artery Disease; CABG, Coronary Artery Bypass Graft; ICE, Intracardiac Echocardiography; LVEF, left ventricular ejection fraction; TAVR, Transaortic Valve Replacement; TOE: Transesophageal Echocardiography; OAC, Oral Anticoagulant.

**Table 2 jcm-10-03443-t002:** Procedural and post-procedural characteristics (according to the presence or absence of reverse atrial remodeling).

	RAR Group *N* = 6	No RAR Group *N* = 37	*p*
Procedure			
Device success	6 (100)	37 (100)	1.00
Technical success	6 (100)	36 (97)	0.875
Procedural success	6 (100)	35 (95)	0.758
Total time in the lab, min	81 ± 25	94 ± 33	0.346
Fluoroscopy time, min	30 ± 18	23 ± 13	0.407
Contrast used (mL)	57 ± 19	59 ± 34	0.868
Mean number of devices used	1.2 ± 0.5	1.2 ± 0.4	0.522
Size of the device used	21 ± 4	23 ± 4	0.507
In-Hospital MACE			
Vascular complication	0 (0)	0 (0)	1.00
Cardiac tamponnade	0 (0)	1 (3)	0.860
Device migration	0 (0)	1 (3)	0.860
Device thrombus	1 (16)	3 (8)	0.465
Stroke/TIA	0 (0)	0 (0)	1.00
Post-procedure CT			
Time delay CT_1_-LAAO	27 ± 41	40 ± 44	0.502
Time delay LAAO-CT_2_	50 ± 16	48 ± 38	0.839
Left atrial volume CT_1_	127 ± 46	141 ± 66	0.551
Left atrial volume CT_2_	151 ± 51	138 ± 64	0.597
Device thrombus	0	0	1.00
No peridevice leak	6 (100)	33 (89)	0.349
Small peridevice leak	0 (0)	3 (8)	0.630
Large peridevice leak	0 (0)	1 (3)	0.860
MACE at 1Y-FU			
FU in days	801 ± 474	736 ± 383	0.761
Stroke/TIA	0 (0)	0 (0)	1.00
Cardiac death	0 (0)	2 (6)	0.381
Non cardiac death	0 (0)	4 (11)	0.063
Major bleeding	1 (16)	3 (8)	0.465
Device migration	0	1 (3)	0.860
Heart failure	0 (0)	3 (8)	0.622

Data are presented as *n* (%) or Mean ± SD.

**Table 3 jcm-10-03443-t003:** Pre-procedural and post-procedural characteristics on CT.

	Pre-Procedure CT	Post-Procedure CT
Time delay CT-LAAO (day)	39 ± 43	48 ± 36
Left atrial volume CT (mL)	139 ± 64	140 ± 63
Reverse remodeling (LA)	-	6 (14%)

Data are presented as *n* (%) or Mean ± SD.

## Data Availability

The data that support the findings of this study are available on request from the corresponding author. The data are not publicly available due to privacy or ethical restrictions.

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
