# Peer review of "Left Atrial Remodeling and Brain Natriuretic Peptide Levels Variation after Left Atrial Appendage Occlusion"

_jcm, 2021, doi:10.3390/jcm10153443_

Round 1
Reviewer 1 Report
Enjoyed reading your interesting study please consider the following comments and suggestions:
-Please verify data in Table 1 (a value of NT probnp 2987+6766 pg/ml in the No RAR patients seems to high)
- Please underscore the news finding of the study and the novelity respect the existing literature
Author Response
I thank the reviewer for this pertinent comment.
Concerning Table 1 : The result is indeed amazing but after verification, it is correct.
Concerning the new findings : A paragraph has been added to explain the real potential for left atrial remodeling after LAAO, in agreement with previous studies. These hypotheses will have to be confirmed with future randomized controlled trials with regular follow-up by CT scan and also by echocardiography.
Reviewer 2 Report
My understanding is the aim of this study was to analyze the influence of LAAO on BNP level in time (as reflecting the presence of remodeling process), in the comparison to initial value of this substance. Why did not you analyze ANP, which is produced especially in the atria?
The study was properly done even being retrospective one. But the discussion ought to be improved (extended). E.g. the BNP level not only reflects heart failure grade, as you commented, but also the intensity of production of this molecule, for instance. So, there is not only simple relationship between the level of BNP and the grade of heart failure. One ought to consider a few aspects of this matter. I propose to plunge into this problem a little bit more, explaining by the way your choice of BNP, not of ANP in this research. Conclusion: the discussion is relatively modest.
Concerning the english language:
1/ sometimes the language is to "sophisticated": the line 100 and 101 - I propose to use present perfect tense instead of past perfect tense
2/ the line 94: are you sure is correct to write: "from for .." ?
3/ the line 65: I propose "bleeding" instead of "bleed"
4/ the line 79: I propose "increase of cardiac congestion"
5/ the line 186-188: split this sentence into two smaller please - this text is too long or unclearly written as one sentence.
6/ You are using two forms: somewhere" left atrial remodeling" , somewhere "left atrium " - line 19,22. Unify this text please. I am much more convinced to write left atrial (remodeling/appendage).
7/ the line 211-213: adjust tenses, please
8/ the line 201: I propose "..an increase of LV filling presssure"
9/ A technical problem: no space between the text and the number of citation - for instance the line 42.
Author Response
I thank the reviewer for this comment and I agree with that.
Concerning ANP :
A limitation of this study is the lack of dosage of ANP, a specific marker of the left atrial, produced by the atria, which was not available at the start of the study on the blood sampling of the patients. Having data on the ANP would have been better. Concerning production of BNP level : I share this pertinent comment. A paragraph has been added to explain the changes in BNP level which are not necessarily linked to heart failure but also potentially linked to vascular or renal changes, with variation in volemia at the time of the blodd sampling. Concerning the English language : The modifications have been made.